# Generation Mechanism of "Information Cocoons" of Network Users: An Evolutionary Game Approach

Xing Zhang [1], Yongtao Cai [1,*], Mengqiao Zhao [1] and Yan Zhou [2]

1   School of Economics and Management, Zhengzhou University of Light Industry, Zhengzhou 450001, China; zhangxing@zzuli.edu.cn (X.Z.); zmq18737800163@163.com (M.Z.)
2   School of Business Administration, South China University of Technology, Guangzhou 510630, China; zyannier@163.com
*   Correspondence: 17183454648@163.com

**Abstract:** The extensive application of algorithm recommendation technology not only meets the information needs of network users but also leads to the emergence of "information cocoons". On the basis of summarizing three generating mechanisms, namely, the theory of technological innovation, the theory of interest-driven, and the theory of emotional identity, this paper constructs a game model of bilateral evolution between information platforms and network users and simulates the influence path of key factors on the evolution of both parties' main strategies. The research shows that algorithm recommendation technology is the root of "information cocoons" in the algorithm era. As the algorithm technology matures day by day, the cost of using algorithm recommendations on information platforms and the loss cost of accepting algorithm recommendations by network users are constantly decreasing, which causes the information platforms and network users' strategy choice for algorithm recommendation to evolve from {give up and conflict} to {use and accept}, and finally leads to the long-term existence of "information cocoons".

**Keywords:** algorithm era; network users; information cocoons; algorithm recommendation; evolutionary game





## 1. Introduction

"Information Cocoons" was first put forward by Professor Sunstein of Harvard Law School in his book "*Information Utopia—How People Produce Knowledge*" published in 2006. It is used as a metaphor for information dissemination, in which the audience only chooses interesting and pleasing topics from the vast amount of information while rejecting or ignoring other views and contents, just like "cocooning silkworms" [1]. With the wide application of the Internet, the research on information dissemination is constantly enriched. Pariser [2] put forward the phenomena of "filter bubbles" in 2011 and explained it as "The basic code at the heart of the new Internet is pretty simple. The new generation of Internet filters looks at the things you seem to like—the actual things you've done, or the things people like you like—and tries to extrapolate. They are prediction engines, constantly creating and refining a theory of who you are and what you'll do and want next. Together, these engines create a unique universe of information for each of us—what I've come to call a filter bubble—which fundamentally alters the way we encounter ideas and information". For "filter bubbles" and "information cocoons", scholars found "filter bubbles" is a technology for mandatory screening of information received by users, and "information cocoons" is a natural tendency for people to prefer confirmatory information and avoid dissonant information [3]. Therefore, as a technology, "filter bubbles" can be continuously improved in the future technological progress; but as the cognitive biases, "information cocoons" will strengthen the personal tendency of user' information acquisition due to the rapid increase of the scale of network information. It can be seen that compared with "filter bubbles", "information cocoons" will become a more noteworthy social problem in the future.

With the continuous development of artificial intelligence, big data, cloud computing, and other technologies, information has entered the era of intelligent communication [4]; the algorithm has become the soul of communication: in order to realize the capital conversion of traffic, the information platforms take personalized communication, which captures individual information preferences, as its action logic, and widely applies algorithm recommendation technology to all kinds of internet applications, making the information communication change from "homogenization" to "personalization", from "people looking for information" to "information looking for people" [5]. The "information cocoons" brought by the algorithm are a kind of social risk source, which will lead to the information gap between different subjects, narrow the content and ways for network users to obtain information, expand the cognitive differences among groups, aggravate the "individualization" of social cognition, eliminate the "generality" tendency, hinder the formation of public space and social consensus, cause information monopoly, and may further cause serious consequences [6–9]. Therefore, the key to solving the problem of "information cocoons" lies in analyzing the generation mechanism of "information cocoons" under algorithm recommendation, and exploring the general law of the generation of "information cocoons" among network users in the algorithm era from the perspective of the bilateral game between information recommendation platforms and network users, so as to standardize the information dissemination mechanism under algorithm recommendation media, guide the information behavior of network users, and maintain information fairness and social fairness.

For the "information cocoons", although there is a consensus on the harm of "information cocoons" in academic circles, there has always been a dispute about the saying that "algorithm leads to information cocoons". This paper combs the relevant literature; it can be concluded that the emergence of "information cocoons" mainly includes three arguments: technological innovation theory, interest-driven theory, and emotional identification theory.

Innovation theory holds that the innovation and use of algorithm technology lead to the emergence of "information cocoons" [10–12]. The IT revolution has enabled modern media to cater to almost all ideological prejudices and beliefs, giving people the power to customize news and opinions [13,14]. Big data are used for pattern mining, analysis of users' behavior, visualization, and tracking of data [15,16], which created conditions for the partial eclipse of information among network users. Thorson and Wells [17] think that an algorithmic filtering system will push information according to acquired users' behavior data and preference data, which will strengthen users' prior information habits. The blessing of algorithmic technology may amplify users' selective exposure, and strengthen the "framing" effect of information consumption, limiting users' opportunities to contact different opinions [18,19]. Yin [20] pointed out that as a "technology-neutral" algorithm, once abused by algorithm developers or controllers, it may lead to problems such as algorithm discrimination, "information cocoons" and "echo chamber" effect, and algorithm hegemony. Users are driven to make decisions in the "information cocoons" and "filtering bubbles" designed by the algorithm recommendation technology, and the algorithm gradually replaces peoples' position to make intelligent decisions, which gradually leads to the situation that users are confined to the "information cocoons", their decision-making behaviors are controlled, and the data ecology is seriously out of balance [21].

Interest-driven theory pays attention to the influence of interest distribution on the formation of "information cocoons" in the process of information dissemination [22,23]. Out of the pursuit of traffic, the information platforms constantly strengthen the legitimacy of the communication target with network traffic as the ultimate standard by setting complex rules [24]. A personalized recommendation in search engines has increasingly aroused peoples' concerns about the potential negative impact of diversity and public discourse quality. The reduction of information diversity that users are exposed to is usually related to algorithmic filtering of online content and adapting to personal preferences and interests [25], which is an important manifestation of the platforms' pursuit of interests. The recommendation algorithm based on users' interests is widely used in commercial

activities [26,27], such as the e-commerce economy and live broadcasts with goods. It is difficult for algorithm recommendations to eliminate the dilemma that will make users fall into "information cocoons". In the political field, the characteristics of "information cocoons" are also obvious. The selective exposure between the parties to their own demands will lead to irreparable ideological differences between the parties, which will lead to increasingly serious social polarization [28,29].

Emotion theory is mainly based on the satisfaction of users' personalized information and holds that "information cocoons" is the identification of users' personal emotions in a certain sense [30]. Sunstein found that users sometimes even actively search for information contrary to their original position; that is, information selection and information avoidance always appear at the same time. In the environment of information overload, the content people receive will lack differences and tend to focus on similar content [31]. Emotional differences will also affect peoples' reception of information. Anger will strengthen the "echo chamber" in the digital public domain, while fear will reduce this "echo chamber" [32]. To satisfy users' personalized experience, the algorithm has been used by various platforms [33,34], such as academic needs [35], medical needs [36], and social needs [37], while under the recommendation of the platforms, the opinions of users with the same position aggregate with each other, but the visibility of the opposite views is limited [38], which challenges the objectivity of news and narrows the information reception of network users. Zhang [39] pointed out that the reason why algorithm recommendation has become the resistance and interference of social consensus is that the algorithm satisfies the users' preferences infinitely, which makes users and algorithms discipline and shape each other.

From the literature review, it can be found that at present, most scholars' research on the generation mechanism of "information cocoons" is inseparable from the important driving factor of algorithm recommendation technology, and it is basically limited to the case level, mostly focusing on specific cases of an information platform such as Tencent News. The initial meaning of recommendation is to gain insight and grasp of network users and their needs. With the "expansion" of the value core of algorithm technology, it incorporates the contents created by the relationship between individual and environment, individual and group, and individual and society into the calculation logic of the algorithm. Therefore, it is of little practical significance to study whether an information platform has "information cocoons" in a general way. To discuss the causes of "information cocoons", we should judge the effect of "information cocoons" from the angle of interaction between network users and platforms according to the universal law of information movement among subjects, objects, and links in the formation process of this phenomenon. Therefore, this paper builds an evolutionary game model between information platform and network users based on the algorithm and verifies it by Matlab numerical simulation.

## 2. Model Construction and Analysis

### 2.1. Description of Evolutionary Game Subjects

When Sunstein put forward "information cocoons", the concept of algorithm had not been clearly defined. With the maturity of algorithm technology, it is widely used in various content distribution activities [40,41], "information cocoons" can more aptly describe the impact of the algorithm: the algorithm recommendation of the information platform screens and pushes information according to the personal characteristics of network users, and recommends personalized information that highly matches their interests and values to network users [42,43], thus forming a content consumption pattern of "thousands of people and thousands of faces".

Information platforms are media platforms that recommend information for network users and provide services to connect people and information. In the age of universal media [44,45], the mechanism of information platforms' open creation and news recommendation make them have a huge amount of information resources, which is the goal of "traffic is king", mining the historical data of network users to understand users' needs, and making users "put it down" through algorithm recommendation, so as to increase

the total amount of information received and expand the revenue. If the cost of using algorithm recommendation exceeds the platforms' affordability or in some areas because of supervision, the platforms will also give up using algorithm recommendation. Therefore, the strategies set recommended by the information platforms for the algorithm are {use, give up}.

Network users, as receivers of information, have the right to choose the information platforms in information acquisition. Network users will choose the platforms to obtain information according to the degree of satisfaction of their preferences. If the recommended content of the algorithm saves them the loss of uninteresting information or brings satisfaction to their psychological preferences, the strategy of network users is to accept the recommendation of the algorithm. However, once the algorithm recommendation of the information platforms cannot meet their preferences, network users can express their resistance to the algorithm recommendation of the information platforms by changing platforms or stopping receiving information. Therefore, the strategies set adopted by network users for information platforms algorithm recommendation are {accept, conflict}.

*2.2. Basic Assumptions*

According to the basic idea of the evolutionary game model, combined with the behavior characteristics of information platforms and network users, the following basic assumptions are made:

**Assumption 1.** *Without considering the influence of market and policy environment, the information platforms S with network users D are individuals who pursue the maximization of their own payoffs. The information platforms can obtain economic payoffs by increasing the information usage of network users, while network users pursue psychological payoffs such as satisfaction of preferences and pleasure in information acquisition.*

**Assumption 2.** *The use of algorithm recommendation by the information platforms requires certain costs, such as personnel and equipment costs, but due to the differences in the attitudes of the network users, the costs also have certain differences. If the network users accept the algorithm recommendation, the information platforms interact more closely with the information of the network users and the costs incurred by the information platforms, $T_1$, are higher than the costs required when the users hold a resisting attitude, $T_2$. If the network users accept the attractive content recommended by the algorithm of the information platforms, they may suffer from addiction, cognitive dissonance, physical fatigue, and other problems, resulting in additional costs H. The widespread use of algorithm recommendations by information platforms for all users of the network community leads to losses of privacy for users who conflict with algorithm recommendations, which is $R_2$. And the information platforms' behavior infringes on the privacy of the network users and needs to bear the risk, denoted as $R_1$.*

**Assumption 3.** *The information platforms can choose to use or give up the algorithm recommendation, but they cannot predict the attitude of the network users. Whether the platforms choose to give up algorithm recommendation or use algorithm recommendation, as long as the network users do not accept it will cause losses, such as platforms users' loss and users' complaints, which are denoted by $C_1$ and $L_1$, respectively. Acceptance of algorithm recommendations by network users will increase usage and bring additional payoffs for the platforms, which are denoted as $B_1$. The network users who accept algorithm recommendations are willing to provide more privacy to the platforms to obtain better service, so the platforms can legally use the users' privacy data to obtain payoffs V.*

**Assumption 4.** *Network users hold the attitude of accepting or conflicting with the algorithm recommendation of information platforms; if network users accept the algorithm recommendation used by the information platforms, they can obtain the extra payoffs $B_2$, such as the users being able to quickly obtain the information they like. So, if the information platforms give up the algorithm recommendation, it will cause the losses $C_2$ to the users who accept the algorithm recommendation. When the users conflict with the algorithm recommendation, it will cause losses to the users, such as aversion, denoted as $L_2$. See Table 1 for specific parameters and their descriptions:*

**Table 1.** Parameter description.

| Parameter | Description |
|---|---|
| $S$ | Information platforms. |
| $D$ | Network users. |
| $p$ | The probability of information platforms using algorithm recommendation is $p$, and the probability of information platforms giving up algorithm recommendation is $1 - p$. |
| $q$ | The probability of network users accepting algorithm recommendation is $q$, and the probability of network users conflict algorithm recommendation is $1 - q$. |
| $W$ | The net incomes of information platforms give up algorithm recommendation are $W_1$, and the net incomes of network users conflict algorithm recommendation are $W_2$. |
| $B$ | The additional incomes of information platforms using the algorithm recommendation are $B_1$, and the additional incomes of network users who accept algorithm recommendation are $B_2$. |
| $C_1$ | The losses of information platforms giving up algorithm recommendation are $C_1$ (such as users churn and competitiveness decline). |
| $C_2$ | The losses of network users due to information platforms giving up algorithm recommendation (such as information lag and dissatisfaction). |
| $T$ | The costs of information platforms using algorithm recommendation are $T$ (such as wage costs). When network users accept algorithm recommendation, the costs are $T_1$; otherwise, they are $T_2$, $T_1 > T_2$. |
| $L$ | When algorithm recommendation is conflicted, the additional losses of information platforms are $L_1$ (such users complaint) and the losses of network users are $L_2$ (such as aversion). |
| $H$ | Information platforms use algorithm recommendation to attract network users, which will cause additional costs for network users (such as addiction and physical fatigue). |
| $R$ | When information platforms use algorithm recommendation, the privacy losses of network users who conflict algorithm recommendation are $R_2$ and the risk of privacy invasion taken by information platforms is $R_1$. |
| $V$ | The additional incomes of information platforms collect and process the data of network users who accept algorithm recommendation are $V$. |

### 2.3. Model Setting

Based on the above assumptions, referring to [46], inspired by the tragedy of the commons. The evolutionary game profit matrix of information platforms and network users is constructed in the following Table 2.

**Table 2.** Evolutionary game profit matrix of information platforms and network users.

| The Main Strategies | Information Platforms Choose to Use ($p$) | Information Platforms Choose to Give Up ($1 - p$) |
|---|---|---|
| Network users choose to accept ($q$) | $W_2 + B_2 - H$; $W_1 + B_1 + V - T_1$ | $W_2 - C_2$; $W_1 - C_1$ |
| Network users choose to conflict ($1 - q$) | $W_2 - L_2 - R_2$; $W_1 - L_1 - T_2 - R_1$ | $W_2$; $W_1$ |

Based on the above description, we can obtain the expected payoffs of the information platforms choosing to use or abandon the algorithm recommendation strategies $E_1$, $E_2$, and average income $\overline{E_S}$ are

$$E_1 = q(W_1 + B_1 + V - T_1) + (1 - q)(W_1 - L_1 - T_2 - R_1) \tag{1}$$

$$E_2 = q(W_1 - C_1) + (1 - q)W_1 \tag{2}$$

$$\overline{E_S} = pE_1 + (1-p)E_2 \tag{3}$$

The dynamic equation of information platforms' evolutionary game replication is

$$\frac{dp}{dt} = p(E_1 - \overline{E_S})$$
$$= p(1-p)[q(B_1 + C_1 + L_1 + R_1 + V + T_2 - T_1) - (L_1 + T_2 + R_1)] \tag{4}$$

Similarly, the expected payoffs of network users adopting acceptance or conflict strategies for algorithm recommendation $E_1$, $E_2$, and average income $\overline{E_D}$ are

$$E_1 = p(W_2 + B_2 - H) + (1-p)(W_2 - C_2) \tag{5}$$

$$E_2 = p(W_2 - L_2 - R_2) + (1-p)W_2 \tag{6}$$

$$\overline{E_D} = qE_1 + (1-q)E_2 \tag{7}$$

The dynamic equation of network users' evolutionary game replication is

$$\frac{dq}{dt} = q(E_1 - \overline{E_D}) = q(1-q)[p(B_2 - H + R_2 + C_2 + L_2) - C_2] \tag{8}$$

Then, the replicator dynamics can be shown as

$$\begin{cases} \frac{dp}{dt} = p(1-p)[q(B_1 + C_1 + L_1 + R_1 + V + T_2 - T_1) - (L_1 + T_2 + R_1)] \\ \frac{dq}{dt} = q(E_1 - \overline{E_D}) = q(1-q)[p(B_2 - H + R_2 + C_2 + L_2) - C_2] \end{cases} \tag{9}$$

Make $\frac{dp}{dt} = 0$, solve $p = 0$, $p = 1$, $p = \frac{L_1 + T_2 + R_1}{q(B_1 + C_1 + L_1 + R_1 + V + T_2 - T_1)}$, if and only if $p$ is the above result, the probability that the information platforms choose to use the algorithm recommendation strategy is stable; make $\frac{dq}{dt} = 0$, solve $q = 0$, $q = 1$, $q = \frac{C_2}{p(B_2 - H + C_2 + R_2 + L_2)}$, if and only if $q$ is the above result, the probability that the information platforms choose to use the algorithm recommendation strategy is stable; and the following five equilibrium points can be obtained, namely $(0,0)$, $(0,1)$, $(1,0)$, $(1,1)$, $(p^*, q^*)$, $p^* = \frac{L_1 + T_2 + R_1}{q(B_1 + C_1 + L_1 + R_1 + V + T_2 - T_1)}$, $q^* = \frac{C_2}{p(B_2 - H + C_2 + R_2 + L_2)}$.

In order to analyze the stability, a Jacobian matrix composed of Equations (4) and (8) is constructed by referring to Friedman's [47] method of local stability analysis of the Jacobian matrix, which replicates dynamic equations:

$$J = \begin{bmatrix} \frac{\partial}{\partial_p}\left(\frac{dp}{dt}\right) & \frac{\partial}{\partial_q}\left(\frac{dp}{dt}\right) \\ \frac{\partial}{\partial_p}\left(\frac{dq}{dt}\right) & \frac{\partial}{\partial_q}\left(\frac{dq}{dt}\right) \end{bmatrix}$$
$$= \begin{bmatrix} (1-2p)[q(B_1 + C_1 + L_1 + R_1 + T_2 - T_1 + V) - (L_1 + T_2 + R_1)] \\ p(1-p)(B_1 + C_1 + L_1 + R_1 + T_2 - T_1 + V) \\ q(1-q)(B_2 - H + R_2 + C_2 + L_2) \\ (1-2q)[p(B_2 - H + R_2 + C_2 + L_2) - C_2] \end{bmatrix} \tag{10}$$

According to Lyaplov's stability theory, the asymptotic stability of pure strategy equilibrium in asymmetric games can be discussed. It is only necessary to substitute (0,0), (0,1), (1,0), and (1,1) into the Jacobi matrix. Referring to [48], we deal with the replicator equation, and we can obtain an asymptotic stability analysis of local equilibrium points in Table 3.

**Table 3.** Asymptotic stability analysis of local equilibrium points.

| Equilibrium Point | *detJ* | *trJ* | Results |
|---|---|---|---|
| $(0,0)$ | $C_2(L_1 + T_2 + R_1)$ | $-C_2 - (L_1 + T_2 + R_1)$ | It is a stable point under any conditions |
| $(0,1)$ | $C_2(B_1 + C_1 - T_1 + V)$ | $B_1 + C_1 - T_1 + V + C_2$ | It is an unstable point under any conditions |
| $(1,0)$ | $(L_1 + T_2 + R_1) \times$ $(B_2 - H + R_2 + L_2)$ | $(L_1 + T_2 + R_1) +$ $(B_2 - H + R_2 + L_2)$ | It is an unstable point under any conditions |
| $(1,1)$ | $(B_1 + C_1 - T_1 + V) \times$ $(B_2 - H + R_2 + L_2)$ | $-(B_1 + C_1 - T_1 + V)$ $-(B_2 - H + R_2 + L_2)$ | When $T_1 < B_1 + C_1 + V$ and $H < B_2 + L_2 + R_2$ it is a stable point; otherwise, a saddle point or unstable point |

According to the evolutionary game theory, the equilibrium state and evolutionary paths of the above system are divided into the following three situations:

**Situation 1.** *$B_1 + C_1 + L_1 + R_1 + V + T_2 - T_1 < 0$, $B_2 + C_2 + R_2 + L_2 - H < 0$, at this time, the determinant and trace of the Jacobian matrix at each equilibrium point are calculated, and according to the evolutionary stability strategy conditions: $detJ > 0$, $trJ < 0$, the stability of the system is analyzed, and the results are shown in Table 4 below:*

**Table 4.** Stability at each equilibrium point of the system in situation 1.

| Equilibrium | *detJ* | *trJ* | Stability |
|---|---|---|---|
| $(0,0)$ | + | − | *ESS* |
| $(0,1)$ | − | indetermination | instability |
| $(1,0)$ | − | indetermination | instability |
| $(1,1)$ | + | + | instability |
| $(p^*, q^*)$ | 0 | − | saddle point |

**Situation 2.** *$0 < B_1 + C_1 + L_1 + R_1 + V + T_2 - T_1 < L_1 + T_2 + R_1, 0 < B_2 + C_2 + R_2 + L_2 - H < C_2$, calculate the determinant and trace of the Jacobian matrix at each equilibrium point, and the results of stability analysis are shown in Table 5 below:*

**Table 5.** Stability at each equilibrium point of the system in situation 2.

| Equilibrium | *detJ* | *trJ* | Stability |
|---|---|---|---|
| $(0,0)$ | + | − | *ESS* |
| $(0,1)$ | − | indetermination | instability |
| $(1,0)$ | − | indetermination | instability |
| $(1,1)$ | + | + | instability |
| $(p^*, q^*)$ | 0 | indetermination | saddle point |

Combining the results of Tables 3 and 4, Situation 1 and Situation 2 can be combined into $B_1 + C_1 + V < T_1, B_2 + L_2 + R_2 < H$, at this time, (0,1), (1,0), (1,1) both are unstable strategies of system evolution; (0,0) for the evolution and stability strategy of the system, that is, {information platforms gives up, network users conflict}, its phase diagram is shown in Figure 1 below.

**Situation 3.** *$T_1 < B_1 + C_1 + V, H < B_2 + L_2 + R_2$, calculate the determinant and trace of Jacobian matrix at each equilibrium point, and the results of stability analysis are shown in Table 6 below:*

**Table 6.** Stability at each equilibrium point of the system in situation 3.

| Equilibrium | *detJ* | *trJ* | Stability |
|---|---|---|---|
| $(0,0)$ | $+$ | $-$ | *ESS* |
| $(0,1)$ | $+$ | $+$ | instability |
| $(1,0)$ | $+$ | $+$ | instability |
| $(1,1)$ | $+$ | $-$ | *ESS* |
| $(p^*,q^*)$ | $0$ | $+$ | saddle point |

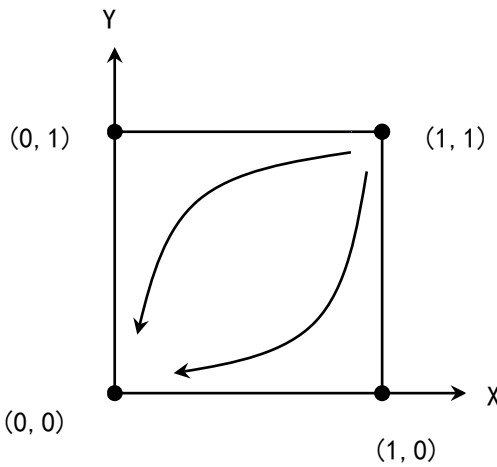

**Figure 1.** Situation 1 and 2 evolution phase diagram.

As can be seen from the results in Table 5, when $T_1 < B_1 + C_1 + V$, $H < B_2 + L_2 + R_2$, (0,1), (1,0) are unstable strategies for system evolution; (0,0), (1,1) for the evolution and stability strategy of the system, that is, {information platform abandons, network users conflict}, {information platform uses, network users accept}, its phase diagram is shown in Figure 2 below.

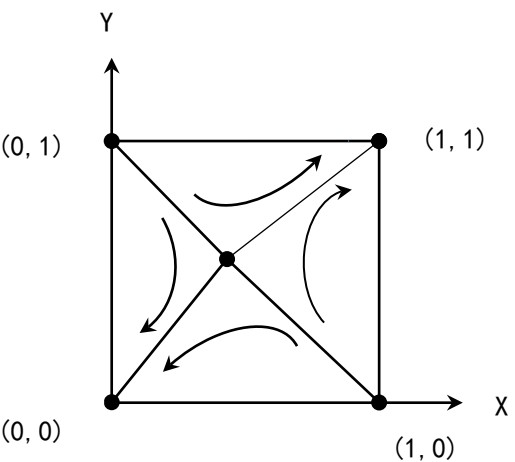

**Figure 2.** Situation 3 evolution phase diagram.

### 2.4. Analysis and Discussion

Based on the above evolutionary game analysis, it is found that the equilibrium result of system evolution may be "information platforms abandon algorithm recommendation; network users conflict with algorithm recommendation" or "information platforms use algorithm recommendation; network users accept algorithm recommendation". The change in the evolutionary equilibrium result is caused by the difference in cost factors. Under the premise of fixed variables such as platforms and network users' extra income,

with the parameters $H$ and $T_1$, in the case of continuous reduction, the system will gradually change from the equilibrium point (0,0) to the balance point (0,0), (1,1) transfer. $B_1 + C_1 + V < T_1$, $B_2 + L_2 + R_2 < H$, under the circumstances, the cost of using algorithm recommendations on information platforms and the losses of network users accepting algorithm recommendation are at a high level, and both the platforms and users are faced with the problem that the cost is higher than the income. $T_1 < B_1 + C_1 + V$, $H < B_2 + L_2 + R_2$, under the circumstances, the cost of the platforms and the loss of users are constantly decreasing, and both parties can maintain a stable profit margin.

By analyzing the path of system evolution combined with market operation, we can find that the result of system evolution accords with the objective law of realistic development. At first, the algorithm recommendation technology just started, and the network users were worried too much about the new algorithm recommendation, and it was difficult to accept it. At the same time, the immature technology led to the high cost of information platforms, so the algorithm recommendation was abandoned by both parties. Subsequently, the development of algorithm technology and its large-scale use, as well as the introduction of relevant rules, have made the rights and interests of network users well protected, and the cost of using algorithm recommendations on information platforms has also been greatly reduced. Therefore, algorithm recommendation can maximize profits for both parties and become the common choice of most information platforms and network users in the algorithm era. However, with the wide application of algorithm technology, there are also individual cases where the acceptance and use of algorithm recommendation by network users and information platforms are still not high (for example, users who are used to books and other information acquisition ways will conflict with algorithm recommendation, and official media such as People's Daily and government platforms will voluntarily give up using algorithm recommendation to ensure the authenticity and comprehensiveness of information).

Based on the above analysis of system stability conditions, we find that "information platforms use algorithm recommendation; network users accept algorithm recommendation" is the stable strategy of the system. Therefore, we believe that the recommendation algorithm maximizes the payoffs of information platforms and network users, and it must exist for a long time and be widely accepted and used by all kinds of information platforms and network users, which will constantly give birth to and strengthen the "information cocoons". The algorithm recommendation of the times deeply participates in peoples' process of acquiring information, constructing their own knowledge system, and understanding the external objective real world, and achieves business goals by reducing the cost of information dissemination, such as reducing the harassment of users by "data garbage", meeting the needs of fragmented reading and accurately targeting the target customers. However, it also shows that information selection behavior may occur in the process of human information acquisition, and this behavior, which originates from human inherent psychology, has been strengthened with the blessing of algorithm technology, leading to the phenomenon of selective approach/avoidance of information, namely "information cocoons".

## 3. Simulation

This paper analyzes the influence of different cost factors on the evolution behavior of information platforms and network users and now simulates the influence of key factors on the evolution of both parties' strategies by Matlab. In reference to the relevant research [49,50], considering that (0,0), (1,1) are the system's ideal equilibrium state, the conditions that must be satisfied are $B_1 + C_1 + V < T_1$, $B_2 + L_2 + R_2 < H$ or $T_1 < B_1 + C_1 + V$, $H < B_2 + L_2 + R_2$. We achieve these conditions by fixing the remaining parameters and changing the values of $T_1$ and $H$, and the values of simulation parameters can be divided into two situations:

(1) The influence of information platforms using algorithm recommendation cost level. The specific assignment of each parameter is shown in Table 7 below:

**Table 7.** The value of each parameter.

| Parameter | $W_1$ | $W_2$ | $B_1$ | $B_2$ | $C_1$ | $C_2$ | $L_1$ | $L_2$ | $H$ | $T_2$ | $R_1$ | $R_2$ | $V$ |
|---|---|---|---|---|---|---|---|---|---|---|---|---|---|
| Value | 10 | 9 | 5 | 3 | 3 | 2.5 | 2 | 5 | 3.5 | 2 | 2 | 4 | 2 |

In this case, there are two possibilities for the information platforms to use the algorithm recommendation cost: lower than or higher than the platform revenue. It can be used to describe two periods of the development of algorithm technology. In the early stage of development, the information platforms need to invest a lot of research and development expenses, so the cost of using algorithm recommendation is extremely high, and even the income is difficult to pay the cost. In the mature period: the mature technology reduces the use cost of algorithm recommendation, and the information platforms can maintain a stable profit margin.

In order to analyze the influence of algorithm recommendation used by information platforms on its strategies selection, the net incomes of information platforms giving up algorithm recommendation strategy and additional incomes increased by using algorithm recommendation are fixed. Then we set different costs $T_1$ to study the probability of information platforms using algorithm recommendation, and the simulation results are shown in Figure 3 below. Ignore the uncertain costs and payoffs, and take the use cost separately $T_1$ for {3,4,18,20}. When $T_1$ for {18,20}, the use cost of algorithm recommendation is higher than the revenue. According to the dynamic evolution process, the probability of using algorithm recommendation by information platforms tends to be 0; the information platforms give up using algorithm recommendation when the cost is high. When $T_1$ for {3,4}, the use cost of algorithm recommendation is lower than the revenue, which can keep a stable profit margin. The probability of using algorithm recommendation for information platforms tends to be 1, so the information platforms choose to use algorithm recommendation when the cost is low.

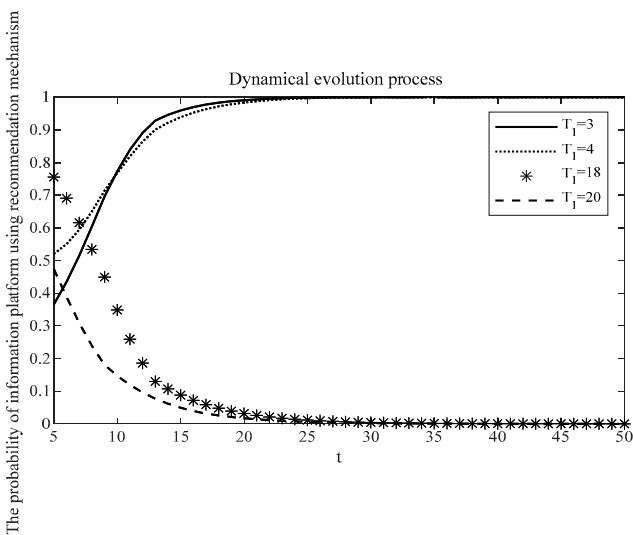

**Figure 3.** Evolution diagram of information platforms strategies under different usage costs.

(2) The network users accept the influence of the loss cost level of algorithm recommendation. The specific assignment of each parameter is shown in Table 8 below.

**Table 8.** The value of each parameter.

| Parameter | $W_1$ | $W_2$ | $B_1$ | $B_2$ | $C_1$ | $C_2$ | $L_1$ | $L_2$ | $T_1$ | $T_2$ | $R_1$ | $R_2$ | $V$ |
|---|---|---|---|---|---|---|---|---|---|---|---|---|---|
| Value | 10 | 9 | 5 | 3 | 3 | 2.5 | 2 | 5 | 5 | 2 | 2 | 4 | 2 |

In this case, there are two possibilities for network users to accept the loss cost of algorithm recommendation: lower than or higher than the users' income. It can be used to describe two periods when network users use algorithm recommendations. At the initial stage of use, network users are resistant to new algorithm recommendation, so the cost of accepting algorithm recommendation is extremely high, and even the gain of network users in recommendation is lower than their loss. At the usage maturity, with the maturity of cognition and the perfection of relevant regulations, the loss possibility of network users accepting algorithm recommendations is getting smaller and smaller, and they can obtain good payoffs.

This paper analyzes the influence of network users' loss cost of accepting algorithm recommendations on their strategy choice. We fix the net income of their conflict with the algorithm recommendation strategy and additional payoffs from accepting the algorithm recommendation. Then, we set different loss costs $H$ to study the probability of users accepting algorithm recommendations, and the simulation results are shown in Figure 4 below. Ignore the uncertain costs and payoffs, and take the loss costs respectively $H$ for {3,714,16}. When $H$ for {14,16}, according to the dynamic evolution process, the probability of network users accepting the algorithm recommendation tends to be 0; when the loss cost is high, network users choose to conflict with algorithm recommendation. When $H$ for {3,7}, network users can maintain a good profit level, so the probability of network users accepting algorithm recommendation tends to be 1. When the loss cost is low, network users choose to accept algorithm recommendations.

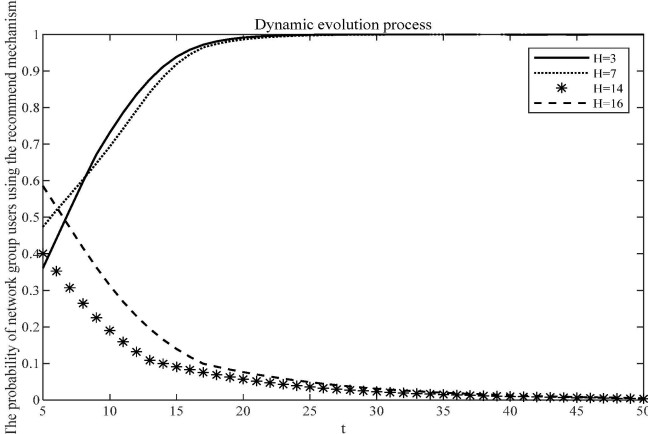

**Figure 4.** Evolution diagram of network users' strategies under different loss costs.

To sum up, the simulation results verify that with the decrease of the recommendation cost of information platforms and the loss of algorithm acceptance by network users, the {give up, conflict} strategy of information platforms and network users towards algorithm recommendation will gradually evolve into the system evolution trend of {use, accept}.

## 4. Conclusions and Managerial Implications

### 4.1. Conclusions

On the basis of summarizing three generating mechanisms, this study found the mechanism of algorithm recommendation for the formation of information cocoons. Therefore, this paper took algorithm recommendation as the core variable to construct a game model of bilateral evolution between information platforms and network users. The study's findings are summarized as follows.

(1) The basic conclusion of this paper: as long as the algorithm recommendation is used, the proliferation of "information cocoons" is inevitable. We found due to the decrease in the cost of using and the loss of network users; algorithm recommendation can maximize the profit of information platforms and network users. Algorithm recommendation must exist for a long time and be widely used and accepted by all kinds of information platforms

and network users, which will continuously give birth to and strengthen the "information cocoons". At the same time, we believe that the maturity and application of algorithm technology have been realized, and the algorithm has become an indispensable tool for network users in information acquisition. Therefore, this "information cocoon" must exist for a long time. As a clustering phenomenon, an increase in the number of information platforms will continue to share the costs of technology investigation and promotion, leading to a decrease in the costs of using algorithm recommendations and a stabilization of the information platforms that use algorithm recommendation; similarly, an increase in size will lead to a stabilization of the number of network users who accept algorithm recommendation. This can be described as a process of adjusting strategies over time and stabilizing in the long term, echoing Ohtsuki and Nowak's research [51].

(2) The strategies of the information platforms for algorithm recommendation are mainly affected by the use costs, and the lower the costs, the greater the probability that the information platforms use the algorithm to recommend. The strategies of network users to algorithm recommendation are mainly affected by the losses, and the smaller the losses of accepting the algorithm recommendation, the higher the probability that the users accept the algorithm recommendation.

### 4.2. Managerial Implications

In view of the social reality of "information cocoons" for a long time, how to avoid or correct the negative effects of "information cocoons", promote social understanding and consensus, and maintain information fairness and social equity has become an important task for information ecological cultivation and social environment governance. Based on the research results, this paper puts forward the following suggestions from the perspective of algorithms, respectively, for technology, information platforms, and network users.

(1) Optimizing algorithm recommendation mechanism. The first step is to optimize the feedback mechanism. We can broaden the interaction field between the information receivers and the algorithm and set up operations such as click, like, screen, and not interested so as to improve the systematicness of user-profiles and realize the optimization of recommendations. The second step is to enrich the information acquisition mechanism, and ensure the information comprehensiveness of the information receivers when using the hot spot, searching, and other functions through the algorithm so as to avoid "all eyes are preference". At the same time, a certain proportion of different types of information is reserved under the channels commonly used by users, such as recommendation and interest partition, to guide users to expand their interests and increase information sources [52,53].

(2) Improving the platform algorithm supervision system. First, we should make overall plans for the algorithm governance of multiple subjects. We should not only pay attention to autonomy but also do a good job in algorithm governance. The information platform should be clear about the awareness of algorithm security and scientific and technological ethics. By employing legal consultants, technical experts, and user representatives, the algorithm supervision center should be set up to achieve multi-agent consultation and algorithm governance so as to realize the prevention and control of algorithm risk at all times so that the algorithm's hidden danger can be dealt with in time. Second, we should refine the classification standards of algorithm supervision. It is necessary to formulate the rights and responsibilities standards for using platform algorithms by integrating the attributes of public opinion, content category quality, user scale, and evaluation to realize the hierarchical classification of algorithm supervision.

(3) Cultivating citizens' awareness of algorithm infringement prevention. Firstly, we should guide citizens to establish a sense of information ecological masters and to enhance the awareness of protecting citizens' information rights and interests through information knowledge publicity to avoid infringement of rights and interests caused by rumors, fraud, and other algorithms. At the same time, it is necessary to crack down on the infringement of big data and privacy theft. Secondly, we should encourage citizens to develop a healthy concept of information consumption. It is necessary to protect the well-being interests of

the public from the application of algorithms and the development of the big data industry, as well as their right to know and trust interests, so as to promote the improvement of algorithm technology.

**Author Contributions:** Conceptualization: X.Z. and Y.C.; formal analysis: X.Z. and Y.C.; funding acquisition: X.Z.; methodology: X.Z. and Y.C.; model construction: Y.C.; project administration: X.Z.; supervision: M.Z.; writing—original draft preparation: Y.C., M.Z. and Y.Z.; writing—review and editing Y.C., M.Z. and Y.Z. All authors have read and agreed to the published version of the manuscript.

**Funding:** This research was funded by China National Social Science Fund (Grant No. 20BTQ045) and Henan Provincial Key Young Teachers Project (Grant No. 2020GGJS127).

**Institutional Review Board Statement:** Not applicable.

**Informed Consent Statement:** Not applicable.

**Data Availability Statement:** Correspondence and requests for materials should be addressed to Yongtao Cai.

**Acknowledgments:** The authors are grateful to the anonymous reviewers for greatly improving the quality of this paper.

**Conflicts of Interest:** The authors declare no conflict of interest.

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
