# Peer review of "Generation Mechanism of “Information Cocoons” of Network Users: An Evolutionary Game Approach"

_systems, doi:10.3390/systems11080414_

Round 1

Reviewer 1 Report

This paper constructs a game model of bilateral evolution between information platforms and network users, and simulates the influence path of key factors on the evolution of both parties’ main strategies. The paper is decently written but could nonetheless be refined as I will explain below.

1. In the part of " 3. Simulation", the value of parameters assignment is not sufficiently referenced. Please provide the basis and explanation.

2. The order of references needs to start at 1.

The paper requires professional native English editing.

Author Response

Dear Teacher:

Thank you very much for your affirmation of our work. We have carefully studied your comments and made corresponding amendments. The specific contents are as follows:

Manuscript ID: systems-2513909;

Title: Generation Mechanism of “Information Cocoons” of Network Users: An Evolutionary Game Approach.

Point 1: In the part of "3. Simulation", the value of parameters assignment is not sufficiently referenced. Please provide the basis and explanation.

Response 1: Thank you for your comment, we previously ignored the detailed description of the parameter assignment and citation, very sorry. We accept your comment and make the following modifications: We have carefully read and cited two articles on the assignment of evolutionary game parameters, and described in detail the conditions for parameter setting (see row 308-312, in red).

Point 2: The order of references needs to start at 1.

Response 2: Thank you for your comment. We have carefully checked and revised the order of references. The current order of references starts from “1”. (see row 28)

Please see the revised paper in attachment.

Reviewer 2 Report

"Generation Mechanism of “Information Cocoons” of Network Users: An Evolutionary Game Approach" is an interesting manuscript, written very clearly and rigorously.
The problem under analysis is very well posed and illustrated through a successful literature review. It seems to me that it is about studying the emergence of a perverse consequence of the dissemination of information, identifying its causes and, consequently, suggesting methods to remedy it.
For this, the authors opt for an evolutionary game approach. Of course, the construction of a model in this context has a lot to do with the subjectivity of analysts, individual opinion of experts, of scholars,... but, in a topic of this type, this seems inevitable to me.
It should be said here that once the proposals have been clearly established, the model seems to me to have been successful. Next are the simulations that constitute the experiments necessary to make the necessary inferences. Here, I would like to know if the ones presented here are examples that characterize many actions carried out, or if they are the only ones carried out, which could prevent the reliability of the conclusions drawn, due to lack of consideration of other relevant situations.
Regarding conclusions, I would like to suggest that this text:

The conclusion of this paper: as long as the basic information technology is used, and the “information cocoons” is inevitable. Due to the decrease...

Should be:

The conclusion of this paper:
-As long as the basic information technology is used, the proliferation of “information cocoons” is inevitable.
Due to the decrease...

I would also suggest checking the writing of the conclusions which, here, have a very trivial air.
In row 28 we have this writing "“cocooning silkworms” [0]." where I don't understand the "[0]".

Author Response

Dear Teacher:

Thank you very much for your valuable comments on ourworks. We have carefully studied your comments and made corresponding amendments. The specific contents are as follows:

Manuscript ID: systems-2513909;

Title: Generation Mechanism of “Information Cocoons” of Network Users: An Evolutionary Game Approach.

Point 1: Here, I would like to know if the ones presented here are examples that characterize many actions carried out, or if they are the only ones carried out, which could prevent the reliability of the conclusions drawn, due to lack of consideration of other relevant situations.

Response 1: Thank you very much for your affirmation of our work. For your question, we explain the following: we fully consider the action characteristics of “network users” and “information platform” in the process of model construction, such as the losses or payoffs they may face in taking each action, and exclude the interaction between each action. For example, we decompose the payoffs by the information platform using the algorithm into two parts: fixed incomes and additional incomes, in order to avoid double counting of payoffs. Therefore, we assume that each action of “network users” and “network platforms” in the paper is independent of each other, and there is no cross relationship between them, which can avoid the confusion of function construction.

Point 2: Regarding conclusions, I would like to suggest that this text: The conclusion of this paper: as long as the basic information technology is used, and the “information cocoons” is inevitable. Due to the decrease... Should be: The conclusion of this paper: -As long as the basic information technology is used, the proliferation of “information cocoons” is inevitable. Due to the decrease... I would also suggest checking the writing of the conclusions which, here, have a very trivial air.

Response 2: Thank you for your comment, we found the confusion in conclusions, very sorry. To this, we have broken down the conclusions and management implications into two parts, “4.1 Conclusions” and “4.2 Managerial Implications”, and supplemented the conclusions (see row 373-385, in orange).

Point 3: In row 28 we have this writing "“cocooning silkworms” [0]." where I don't understand the "[0]".

Response 3: Thank you for your comment. We have carefully checked and revised the order of references. The current order of references starts from “[1]”. (see row 28)

Please see the revised paper attachment

Reviewer 3 Report

The authors use the theory of evolutionary games to explain the mechanism of what they call information cocoons (I do not fully understand how this concept differs from the information bubble, widely discussed in the literature).

In my opinion, the work is not clearly written. For example, the sentence (line 123):”

When Sunstein put forward “information cocoons”, the concept of algorithm had not been clearly defined.”

is incomprehensible to me. The concept of an algorithm has been known for a long time. Maybe the authors had some specific algorithm in mind here.

Understanding the work is hampered by the not very clear explanation of the parameters in the model. This may be due to my little knowledge of network interactions.

This may be due to my little knowledge of network interactions. I think it would be easier for the reader to provide a specific example of behavior in the network (illustrating this mechanism).

There is also a lack of a thorough description of the model in the language of game theory. The set of players, strategies, and payoffs should be clearly written down (the payoffs in table 2 are given in such a way that it is not known which strategy arrangement they correspond to).

Author Response

Dear Teacher:

Thank you very much for your valuable comments on ourworks. We have carefully studied your comments and made corresponding amendments. The specific contents are as follows:

Manuscript ID: systems-2513909;

Title: Generation Mechanism of “Information Cocoons” of Network Users: An Evolutionary Game Approach.

Point 1: The authors use the theory of evolutionary games to explain the mechanism of what they call information cocoons (I do not fully understand how this concept differs from the information bubble, widely discussed in the literature).

Response 1: Thank you for your comment, for your question, we answer the following, I hope you can agree. “Information Cocoon” is a trouble faced by network users in information acquisition. This phenomenon has attracted much attention since Sunstein proposed it in 2006. For the causes of “information cocoon”, most scholars start from the users’ information acquisition behavior, such as questionnaires. Therefore, the conclusions are as follows: firstly, it is believed that technology is the cause of “information cocoon”, because before the widespread adoption of information technology, the users’ information acquisition was mostly based on their own will. But the using of technology make users cannot receive some information, and a more serious “information cocoon” occurs; secondly, it is believed that the interest-driven “information cocoon” is generated. Information platforms hope that network users use their own platform for a long time and high frequency to create more payoffs for themselves. Therefore, information platforms will be keen to use algorithmic to allow users to continuously obtain information they like, resulting in incomplete user information reception; thirdly, the users’ emotional preferences, they are more willing to see their favorite information, which will also lead to incomplete user information reception. We found these three causes, each of which may lead to the users’ “information cocoon”, and these three viewpoints all agree with the effect of algorithm technology on “information cocoon”, but these three viewpoints are based on the fact that algorithm technology has been widely used, so before the emergence and application of information technology? Therefore, this paper argues that the three views are the elaboration of the causes of the “information cocoon”, but they are unilateral. We think about the generation mechanism of the “information cocoon” through evolutionary game, from the emergence of algorithm technology to the process of wide application. For example, we think that the “information cocoon” is not a common anxiety when the algorithm first appears. Obviously, this is different from the content discussed in the literature.

Point 2: In my opinion, the work is not clearly written. For example, the sentence (line 123):” When Sunstein put forward “information cocoons”, the concept of algorithm had not been clearly defined.” is incomprehensible to me. The concept of an algorithm has been known for a long time. Maybe the authors had some specific algorithm in mind here

Response 2: Thank you very much for your comment, your question, we do the following answers. “Information Cocoon” was first proposed by Sunstein in the 2006 publication of 《Information Utopia - How People Produce Knowledge》, but the book does not describe the concept of “Information Cocoon” in detail, only as a social phenomenon. What we want to describe in line 123 is “when first proposed, the concept of ‘information cocoon’ is not specific”, which may be difficult for you to understand, we are very sorry. We also very much agree with your point, due to the deepening of research, the current concept of “information cocoon” has been very clear. Algorithms are also common in our daily information acquisition, such as deep learning algorithms and user portrait technology, which meet our needs for obtaining favorite information, but also make us gradually not get used to receiving other types of information, which is worth thinking about.

Point 3: Understanding the work is hampered by the not very clear explanation of the parameters in the model. This may be due to my little knowledge of network interactions. This may be due to my little knowledge of network interactions. I think it would be easier for the reader to provide a specific example of behavior in the network (illustrating this mechanism).

 Response 3: Thank you very much for your comment, we found the confusion for the parameter description, very sorry. To this, we have systematically modified the parameter descriptions in Table 1 (see row 188, in green).

Point 4: There is also a lack of a thorough description of the model in the language of game theory. The set of players, strategies, and payoffs should be clearly written down (the payoffs in table 2 are given in such a way that it is not known which strategy arrangement they correspond to).

Response 4: Thank you very much for your comment, we have adjusted it. For the indistinct writing of game strategy, we have made a mark, you can see the strategy set of network users and information platform (row 139-141 and row 150-152, in green); for the table 2, we have improved (see row 192, in green ). For the two benefits in the cells, the upper are the network users’ payoffs, and the lower are the information platforms’ payoffs. For this writing method, we have read other literatures and found that this is a more common writing method.

Please see the revised paper 

Reviewer 4 Report

This paper investigates the “information cocoons” phenomenon arising from the widespread adoption of algorithmic recommendation technology. Drawing on three theoretical frameworks - technological innovation theory, interest-driven theory, and emotional identity theory - a game model is developed to analyze the dynamic evolution between information platforms and network users. Through simulations that capture the influence of key factors on the strategies of both parties, the study uncovers that algorithmic recommendation technology plays a pivotal role in the formation of "information cocoons" in the algorithmic era. With the continuous advancement of algorithm technology, the costs associated with using algorithmic recommendations on information platforms and the potential losses experienced by network users diminish gradually. Consequently, the strategic choices of information platforms and network users shift from a state of "give up and conflict" to "use and accept," perpetuating the persistence of "information cocoons" over the long term.

Page 4, Table 1. Not clear at all. Please reformulate the descriptions of the parameters.

On page 5, Table 2, your game reminds me of a dynamic game of invasion and resistance from Dragicevic (2015) (Bayesian Population Dynamics of Spreading Species, Environmental Modeling and Assessment) which was based on a balance equation.

On pages 5-6, Your stability study is a little bit blurry. For a peer-reviewed comparative study, take a look at Dragicevic (2019) (Reflective Evolution under Strategic Uncertainty, International Journal of Bifurcation and Chaos) which also deals with the replicator equation.

General comment: I would not speak of benefits (or expected benefits) but of payoffs (or expected payoffs), since you do not have profit functions.

On page 8, lines 281-293, where does your analysis originate? Is it derived from the framework of evolutionary game theory? If so, it is necessary to elaborate on the methodology employed to arrive at your conclusions. Alternatively, if your analysis is not based on evolutionary game theory, please provide the source of your assertions.

Page 10, lines 358-359: “The conclusion of this paper: as long as the basic information technology is used, and the “information cocoons” is inevitable.” I do not understand the meaning of this sentence. Could you please rephrase it?

In the conclusive section, you could discuss the possibility of modeling the replicator dynamics on graphs, with the appearance of information cocoons as clusters within the networks. Please take a look at Ohtsuki and Nowak (2006) (The Replicator Equation on Graphs, Journal of Theoretical Biology).

Could be improved.

Author Response

Dear Teacher:

Thank you very much for your valuable comments on ourworks. We have carefully studied your comments and made corresponding amendments. The specific contents are as follows:

Manuscript ID: systems-2513909;

Title: Generation Mechanism of “Information Cocoons” of Network Users: An Evolutionary Game Approach.

Point 1: Page 4, Table 1. Not clear at all. Please reformulate the descriptions of the parameters.

Response 1: Thank you very much for your comment, we found the confusion for the parameter description, very sorry. To this, we have systematically modified the parameter descriptions in Table 1 (see row 188, in green).

Point 2: On page 5, Table 2, your game reminds me of a dynamic game of invasion and resistance from Dragicevic (2015) (Bayesian Population Dynamics of Spreading Species, Environmental Modeling and Assessment) which was based on a balance equation.

Response 2: We very much recognize your opinion that evolutionary game is to analyze their respective strategies by constructing the payoff matrix of the subject. The dynamic game of invasion and resistance of Dragicevic (2015) (Bayesian Population Dynamics of Spreading Species, Environmental Modeling and Evaluation) can be used as a classic work of evolutionary game, which is worthy of our serious study.

Point 3: On pages 5-6, Your stability study is a little bit blurry. For a peer-reviewed comparative study, take a look at Dragicevic (2019) (Reflective Evolution under Strategic Uncertainty, International Journal of Bifurcation and Chaos) which also deals with the replicator equation.

Response 3: We are very sorry for ignoring the detailed analysis of the stability of the system. To this, we add the asymptotic stability analysis of the local equilibrium points in the paper (see table 3, row 223-228, in purple).

Point 4: General comment: I would not speak of benefits (or expected benefits) but of payoffs (or expected payoffs), since you do not have profit functions.

Response 4: Thank you very much for your comment, we have adjusted the “benefits” (or “expected benefits”) in the text (see row 159, 161, 178, 180, 183, 193, 202, 292, 330, 352, 355,358, in purple)

Point 5: On page 8, lines 281-293, where does your analysis originate? Is it derived from the framework of evolutionary game theory? If so, it is necessary to elaborate on the methodology employed to arrive at your conclusions. Alternatively, if your analysis is not based on evolutionary game theory, please provide the source of your assertions.

Response 5: Thank you very much for your comment. We hope to briefly summarize the above conclusions through this section, so that readers can more clearly understand that the above evolutionary game can prove the role of algorithm recommendation in the generation mechanism of “information cocoons”. To this, we elaborate on the method used before the paragraph (see row 289-291, in purple).

Point 6: Page 10, lines 358-359: “The conclusion of this paper: as long as the basic information technology is used, and the “information cocoons” is inevitable.” I do not understand the meaning of this sentence. Could you please rephrase it?

Response 6: Thank you very much for your comment, we have replaced “The conclusion of this paper: as long as the basic information technology is used, and the ‘information cocoons’ is inevitable.” with “The basic conclusion of this paper: as long as the algorithm recommendation is used, the proliferation of ‘information cocoons’ is inevitable”. We proved the effect of algorithm technology on the generation of “information cocoons” through the model, that is, the algorithm contributed to the “information cocoons”, our previous expression is not clear, I hope you can recognize this modification. (see row 374-375, in orange )

Point 7: In the conclusive section, you could discuss the possibility of modeling the replicator dynamics on graphs, with the appearance of information cocoons as clusters within the networks. Please take a look at Ohtsuki and Nowak (2006) (The Replicator Equation on Graphs, Journal of Theoretical Biology).

Response 7: Thank you very much for your commet, we believe that this can be used as an important direction for our next research, “information cocoon” has obvious group characteristics, in-depth research can make us more systematic understanding of this group phenomenon. However, considering that our understanding of the research methods of Ohtsuki and Nowak (2006) (The Replicator Equation on Graphs, Journal of Theoretical Biology) is not deep, and we find that the research of this paper does not involve more “replicator dynamics on graphs” related research, we will seriously consider this problem in further research.

Please see the revised paper

Round 2

Reviewer 3 Report

However, I would encourage the authors to explain to the readers (in the text) in simple terms what is the difference between an information bubble and an information cocoon. The more so that they cite works on information bubbles in the bibliography. However, the word bubble does not appear in the text. On the plus side, it should be noted that the authors have slightly improved the readability of the work. However, they must also take care to eliminate editorial errors (e.g. caption under Figure 1).

Reviewer 4 Report

Thank you for providing the updates. I have thoroughly reviewed the revised manuscript, and I would like to express my appreciation for the efforts made by the authors in addressing the reviewer's concerns. However, I must mention that there are still areas that require improvement. Firstly, Table 1 remains challenging to read, and I believe it could benefit from further refinement to enhance its clarity. Secondly, I observed that none of the suggested literature has been taken into account to demonstrate that the authors' approach has already been tested. Additionally, in my opinion, overlooking the possibility that information cocoons could be network clusters represents a flaw that should be considered. Lastly, I find the conclusive section to be unsatisfactory, and I recommend further revisions to enhance its effectiveness in summarizing the key findings and implications of the study. Considering these points, I strongly suggest a second round of revision to address these issues and elevate the manuscript's overall quality. However, I acknowledge that the final decision rests with the editor.

Could be improved.

Round 3

Reviewer 4 Report

Thank you for the updates.

I have nothing to add.